

# Combining ability, heterosis and performance of grain yield and content of Fe, Zn and protein in bread wheat under normal and late sowing conditions

Gita R. Chaudhari[1], D. A. Patel[1], D. J. Parmar[2], K. C. Patel[3] and Sushil Kumar[4]

[1] Department of Genetics & Plant Breeding, BACA, Anand Agricultural University, Anand, Gujarat, India
[2] Department of Statistics, BACA, Anand Agricultural University, Anand, Gujarat, India
[3] Micronutrient Research Centre (ICAR), BACA, Anand Agricultural University, Anand, Gujarat, India
[4] Department of Agricultural Biotechnology, Anand Agricultural University, Anand, Gujarat, India

Corresponding author
Gita R. Chaudhari,
geetchaudhary89@gmail.com

## ABSTRACT

Wheat (*Triticum aestivum* L.) is inherently low in protein content, Zn and Fe. Boost yield gains have unwittingly reduced grain Zn and Fe, which has had negative impacts on human health. The aim of this study was to understand the inheritance of grain yield per plant and grain Fe, Zn, and protein concentrations in bread wheat (*Triticum aestivum* L.) under normal and late sown conditions. Half diallel crosses were performed using 10 parents. The crosses and parents were evaluated in replicated trials for the two conditions, to assess the possibility of exploiting heterosis to improve micronutrient contents. The *per se* performance, heterosis, combining ability, and genetic components were estimated for different characters in both environments. The results revealed that hybrid GW 451 × GW 173 exhibited better parent heterosis (BPH) and standard heterotic effects (SH) in all environments. In both sowing conditions, the general combining ability (GCA) effects of poor × poor parents also showed high specific combining ability (SCA) effects of hybrids for both the micronutrients and protein contents. However, $\sigma^2 A/\sigma^2 D$ greater than unity confirmed the preponderance of additive gene action for protein content, and GW 173 was identified as a good general combiner for these characteristics under both environments. SCA had positive significant ($P < 0.001$) correlations with BPH, SH1, SH2, and the phenotype for yield component traits and grain protein, Fe, and Zn concentrations in both conditions. A supplementary approach for biofortifying wheat grainis required to prevent malnutrition.

## INTRODUCTION

Wheat (*Triticum aestivum* L.) is a crucial staple food crop worldwide, and has been consumed since ancient times by human beings (*Giraldo et al., 2019*). Worldwide, wheat is cultivated on more than 17% of all cultivable land, occupying approximately 32% of the

total land under cereal cultivation. It is consumed by nearly 40% of the global population, and fulfills 21% of the daily protein requirements of more than 4.5 billion people in developing countries (*Giraldo et al., 2019*). Micronutrient malnutrition is a serious public health concern in several parts of the world, especially underdeveloped countries (*Ragi et al., 2021*). Hidden hunger, or micronutrient deficiency is one of major reasons for anemia. Currently, 60% and 30% population is deficient in Fe and Zn, respectively (*Ragi et al., 2021*). In India, 80% of pregnant women, 52% of non-pregnant women, and 74% of children aged approximately 6–35 months suffer from induced anemia-Fe deficiency (IDA) (*Sharma et al., 2021*). Biofortification can be a useful tool to fight this "hidden hunger," playing a crucial role in the improvement of micronutrient contents in the diet, which will benefit billions of people (*Bouis & Saltzman, 2017*). Micronutrient deficiencies in humans could be altered through biofortification, mineral supplementation, and food diversification and fortification (*Gupta, Hossain & Muthusamy, 2015*). Biofortification is the action of increasing the content of micronutrients in the edible parts of staple food crops to ensure better human nutrition (*Prasanna et al., 2020*). However, the root cause of micronutrient malnutrition is staple crop-based diets that generally do not furnish the required amounts of heavy assurance and micronutrients (*Wakeel et al., 2018*). The use of biofortified cultivars is needed to alleviate micronutrient malnourishment in large populations, especially in Africa (*Sofield et al., 1977*). Biofortification strategies cover breeding in wheat, genetic control, and utilization of mineral fertilizers, and have broad potential to address micronutrient malnourishment (*Garg et al., 2018*).

Different sowing dates are primarily connected to the effects of temperature and day duration (*Chaudhari, 2022*). Low wheat yields in central India are caused by the short winter season and high temperatures throughout the growing season. Here, the main effects of high temperature are decreased grain weight and reduced grain filling duration (*Wardlaw et al., 1989*; *Mutwali et al., 2015*). Escape mechanism is functional in late sown wheat to reduce the effects of high temperature during grain filling while early sown wheat use escape mechanism to cope with problem of high temperature during germination and vegetative growth. Consequently, recognition of the best wheat genotypes appropriate for particular sowing conditions is of considerable significance to attain higher yields and micronutrient uptake. Understanding associations between grain yield, protein, and micronutrient content under normal and late sown conditions will be helpful for heterosis breeding/hybrid wheat technology. Micronutrient supplementation provides a balanced nutritional diet for people who suffer from deficiencies due to dependence on a wheat grain diet for their Fe and Zn intake. Therefore, to tackle micronutrient insufficiency among wheat eaters, attempts must be made to increase micronutrients, especially grain Fe and Zn content accompanied by supportable crop grain yield. The objective of this study was to evaluate the *per se* performance, heterosis, and combining ability of wheat parental lines and the hybrids produced from them in terms of grain Fe and Zn content under normal and late seeded conditions. These results directly affect the efficacy of producing high-yielding hybrids with high levels of Fe and Zn densities under different environment conditions.

## MATERIALS AND METHODS

### Plant material and field performance evaluation

The experimental material consisted of 55 entries comprising 45 hybrids generated by following a diallel mating design (*Hayman, 1954*, *1957*, *1960*) using 10 parental genotypes and excluding reciprocal crosses with two standard check varieties, MACS 6222—TS and HD 2932—LS. The details of the parental genotypes and checks and hybrid crosses are given in Tables S1 and S2. The represented genotypes were cultivated (commercial varieties and checks). All the experiments were performed in accordance with relevant guidelines and regulations. All plant materials were available within the institute and are available in the public domain, eliminating the need to obtain permissions. The crosses were made during *rabi (*an Indian term for the winter season) 2017–18 in 10 × 10 diallel mating, excluding reciprocals in Table S1.

The parents, hybrids, and standard checks were arranged in a randomized complete block design (RCBD) with three replications at the Regional Research Station, Anand Agricultural University, Anand, India (22°35′N, 72°55′E) with two diverse crop sowing methods, *i.e.*, normal ($E_1$) and late ($E_2$) conditions during *Rabi* 2018–19. Each treatment consisted of a single row 3.0 m in length with 20 cm inter row spacing and 10 cm intra row spacing for normal ($E_1$) conditions. These measurements were 18 and 10 cm respectively for late ($E_{2)}$ conditions. The crops were raised following the recommended practices in Table S3. Immediately after sowing, need-based supplementary irrigation was provided in Table S3. The data were recorded for grain yield per plant (GYP), protein content (PC), Fe (Iron) and Zn (Zinc) content in grain. The data for the studied traits in parents and hybrids were subjected to an analysis of variance (ANOVA) using a general linear model as implemented in using SPAR version 2.0 (*Ahuja et al., 2008*). The soil of the experimental site is sandy loam, alluvial in origin, deep, well drained, and had fairly good moisture holding capacity. This is typically "Goradu" soil, prevalent in the Charotar region of Gujarat. This area has a tropical and semi-arid climate. Weather data were recorded at the Observatory, Department of Meteorology, B. A. College of Agriculture, A.A.U., Anand.

**Grain protein.** These parameters were estimated using Fourier transform-near infrared reflectance spectroscopy (FT-NIRS) at the Centre of Excellence for Research on Wheat, Sardar Krushinagar Dantiwada Agricultural University, Vijapur. FT-NIRS is a rapid and non-destructive technique, which is largely used in the food and feed industry.

**Grain Fe and Zn analysis.** All samples of wheat grains were analyzed to determine the contents of micronutrients (Fe and Zn) using inductively coupled plasma optical emission spectroscopy (ICP-OES), Model Optima 7,000 DV, Perkin Elmer, USA, and the reliability of the data was ensured by analyzing blanks at the Micronutrient Research Center, Anand Agricultural University, Anand.

### Estimation of genetic parameters

Subsequently, the genetic parameters, including the variance due to general combining ability ($\sigma^2$GCA), variance due to specific combining ability ($\sigma^2$SCA), additive genetic variance ($\sigma^2$A), dominance genetic variance ($\sigma^2$D), and ratio of additive and dominance

**Table 1 Mean squares and genetic components for GYP, PC, grain Fe and Zn frequency in a half diallel design under normal and late conditions.**

| Source | Df | Normal | | | | Late | | | |
|---|---|---|---|---|---|---|---|---|---|
| | | GYP | PC | Fe | Zn | GYP | PC | Fe | Zn |
| Replications | 2 | 2.38 | 0.01 | 0.65 | 0.05 | 9.48 | 0.25 | 0.10 | 0.16 |
| Genotypes | 56 | 57.87** | 1.67** | 37.51** | 4.88** | 89.86** | 0.56** | 31.36** | 5.92** |
| Parents | 9 | 26.56** | 2.33** | 49.72** | 4.51** | 71.70** | 0.58** | 32.59** | 1.46** |
| Hybrids | 44 | 67.93** | 0.78** | 33.27** | 4.86** | 82.67** | 0.51** | 32.36** | 4.38** |
| Parents v/s Hybrids | 1 | 7.35 | 31.90** | 11.77** | 18.63** | 485.7** | 1.73** | 7.99** | 93.52** |
| Check v/s Hybrids | 1 | 2.14 | 0.005 | 91.09** | 0.07 | 262.76** | 1.42** | 33.14** | 40.63** |
| Bet. checks | 1 | 3.68 | 5.73** | 79.62** | 0.03 | 58.28** | 0.68* | 0.54 | 1.48** |
| Error | 112 | 3.35 | 0.01 | 0.65 | 0.14 | 4.19 | 0.13 | 0.89 | 0.21 |
| Proportional contribution of parents and hybrids | | | | | | | | | |
| Replications | 2 | 3.00 | 0.011 | 0.88 | 0.050 | 7.64 | 0.23 | 0.05 | 0.30 |
| Parents | 9 | 48.88** | 6.56** | 10.55** | 3.51** | 25.58** | 0.25** | 6.71** | 2.59** |
| Hybrids | 45 | 14.19** | 0.51** | 12.13** | 1.32** | 30.20** | 0.16** | 11.43** | 1.70** |
| Genetic components | | | | | | | | | |
| $\sigma^2$GCA | | 2.89** | 0.50** | −0.13 | 0.18** | −0.39 | 0.01 | −0.39 | 0.07 |
| $\sigma^2$SCA | | 13.09** | 0.51** | 11.93** | 1.28** | 28.80** | 0.12** | 11.13** | 1.64** |
| $\sigma^2$A | | 5.78 | 1.01 | −0.26 | 0.37 | −0.77 | 0.02 | −0.79 | 0.15 |
| $\sigma^2$D | | 13.09 | 0.51 | 11.93 | 1.28 | 28.80 | 0.12 | 11.13 | 1.64 |
| $(\sigma^2 A/\sigma^2 D)^{0.5}$ | | 0.44 | 1.98 | 0.021 | 0.28 | 0.026 | 0.16 | 0.070 | 0.09 |

Note:

df, degrees of freedom; *P < 0.05; **P < 0.01. GYP, grain yield per plant; PC, protein content; Fe, grain Fe content; Zn, grain Zn content.

genetic variance $((\sigma^2 A/\sigma^2 D)\ 0.5)$ were estimated. These genetic components of variance were calculated from the ANOVA table (*Griffing, 1956*).

## RESULTS

### ANOVA performance

In Table 1, the performance of hybrid and parental lines is presented. The mean squares of genotypic effects for the traits under study were significant at *P* < 0.01 in both sowing conditions.

In Table 1, the outlined diallel mating design is analyzed using ANOVA. The ANOVA for combining ability in $E_1$ and $E_2$ environments revealed that mean squares of parents and hybrids were significant (*P* < 0.01) for all characters.

The estimates of $\sigma^2_{GCA}$ and $\sigma^2_{SCA}$ revealed that additive and non-additive gene effects were involved in inheritance of GYP and Zn content in $E_1$. Additionally, the estimates of $\sigma^2_{SCA}$ were significant for all characters in both environments, which suggested the importance of only the non-additive gene effects for inheritance of these characters.

The value of the average degree of dominance was less than one for the characters GYP in $E_1$, protein content in $E_2$ and zinc content in $E_1$. Partial dominance behavior was revealed by the interaction of alleles for these characters. For the Fe content in $E_1$ and $E_2$

and zinc content in $E_2$, the value of the average degree of dominance was close to zero or zero, revealing an absence of dominance for both characters.

## Hybrids and parental lines performance

The *per se* performance of parental lines, hybrids, and checks are presented in Table 2. In Table 2, coefficient of variation (CV) <10% for the calculated traits indicated acceptable and efficacious experiment results.

The GYP in hybrids ranged from 10.0 g (H45) to 32.9 g (H3) in $E_1$ and from 9.4 g (H27) to 35.4 g (H24) in $E_2$. The standard check HD 2932 had the highest GYP compared with MACS 6222 in both environments. The PC in hybrids ranged from 9.9% to 11.7% in $E_1$ and 10.6% to 12.4% in $E_2$. In case of standard checks, MASC 6222 had a higher protein content compared to HD 2932 in both environments.

The Fe content hybrids exhibited variation from 21.7 ppm (H2) to 41.5 ppm (H14) in $E_1$ and from 22.3 ppm (H34) to 36.0 (H37) in $E_2$. In the standard check, MACS 6222 recorded higher Fe content compared to HD 2932 in both environments. The hybrids, H36 (15.4 ppm) and H14 (14.1 ppm) in $E_1$ and H45 (14.6 ppm) in $E_2$ had higher Zn contents. The standard check MACS 6222 had the highest Zn content compared with HD 2932 in both environments.

## Heterosis performance

Two categories of heterosis, standard heterosis (SH) and better parent heterosis (BPH), exhibited significant variation under both sowing environments (Tables 3 and 4). In this study, several crosses indicated noticeable heterotic responses over better parent (heterobeltosis) and standard check varieties (standard heterosis). These were MACS 6222 for normal sown and HD 2932 for late sown.

For GYP, four $F_1$ progenies had significant positive estimates under both environments. Between these, H3 and H6 had higher BPH under normal conditions. Overall, H6 had significant positive estimates for BPH (normal, 52.69%; late, 17.41%). H3 and H24 had the highest positive SH for $E_1$ (99.41%) and $E_2$ (65.64%).

However, the higher percentage of hybrids showed a more positive BPH under late than normal conditions for PC. One $F_1$ progeny, H30, under normal conditions, and eight $F_1$ progenies under late conditions showed significant positive BPH. In particular, under the late condition there were higher significant positive crosses for SH than under the normal condition.

For Fe content, H5, H9, and H36 had consistently significant positive heterosis for BPH under both conditions. H5, H7, H8, H28, H30, H32, and H36 were significantly positive for SH under both checks and environments. Interestingly, H5 had consistently significant positive BPH and SH estimates for Fe content under both environments, and GYP under normal conditions (SH).

For Zn content, H45 had consistently significant positive BPH under both environments and significant positive SH over both checks, except $SH_1$ in $E_1$. In addition, H36 had the highest significant positive estimates of SH for Zn content under normal

**Table 2 Performance of F₁ progenies (hybrids), parent lines, and checks for GYP, PC, grain Fe and Zn content under normal and late conditions.**

| Entries | Normal | | | | Late | | | |
|---|---|---|---|---|---|---|---|---|
| | GYP (g) | PC (%) | Fe (ppm) | Zn (ppm) | GYP (g) | PC (%) | Fe (ppm) | Zn (ppm) |
| Parents | | | | | | | | |
| GW 451 (P1) | 15.0 | 11.5 | 22.2 | 11.5 | 20.3 | 10.4 | 24.8 | 14.9 |
| GW 496 (P2) | 14.2 | 11.2 | 28.7 | 11.1 | 21.1 | 10.5 | 26.3 | 14.0 |
| LOK 1 (P3) | 21.7 | 12.1 | 27.8 | 11.5 | 26.8 | 11.3 | 28.2 | 12.7 |
| GW 322 (P4) | 17.2 | 11.4 | 23.6 | 11.5 | 24.4 | 10.8 | 24.2 | 12.9 |
| GW 366 (P5) | 17.3 | 10.9 | 22.9 | 10.9 | 32.2 | 10.5 | 33.3 | 13.4 |
| HI 1544 (P6) | 23.1 | 11.9 | 25.4 | 13.9 | 24.7 | 11.3 | 27.3 | 12.5 |
| GW 173 (P7) | 14.5 | 13.4 | 28.7 | 12.9 | 19.0 | 11.8 | 31.3 | 13.3 |
| GW 11 (P8) | 16.6 | 12.9 | 31.4 | 11.8 | 16.7 | 11.3 | 28.8 | 13.7 |
| HD 2864 (P9) | 16.5 | 12.2 | 34.4 | 10.7 | 20.1 | 10.9 | 33.2 | 13.6 |
| UAS 385 (P10) | 15.6 | 10.6 | 23.2 | 9.4 | 16.4 | 10.9 | 31.3 | 13.7 |
| Mean | 17.2 | 11.8 | 26.8 | 11.5 | 22.2 | 11.0 | 28.8 | 13.5 |
| Min. | 14.2 | 10.6 | 22.2 | 9.4 | 16.4 | 10.4 | 24.2 | 12.5 |
| Max. | 23.1 | 13.4 | 34.4 | 13.9 | 32.2 | 11.8 | 33.3 | 14.9 |
| Hybrids | | | | | | | | |
| GW 451 × GW 496 (H1) | 20.0 | 10.7 | 25.9 | 11.2 | 16.1 | 11.5 | 27.3 | 10.4 |
| GW 451 × LOK 1 (H2) | 29.8 | 10.4 | 21.7 | 9.8 | 20.2 | 11.9 | 29.6 | 10.5 |
| GW 451 × GW 322 (H3) | 32.9 | 10.2 | 24.4 | 9.3 | 23.0 | 11.9 | 33.5 | 11.1 |
| GW 451 × GW 366 (H4) | 15.8 | 10.2 | 24.2 | 8.8 | 16.4 | 11.6 | 30.1 | 11.0 |
| GW 451 × HI 1544 (H5) | 20.4 | 10.0 | 29.1 | 9.5 | 18.1 | 12.3 | 32.2 | 10.9 |
| GW 451 × GW 173 (H6) | 22.9 | 10.1 | 23.2 | 9.8 | 23.8 | 11.5 | 31.0 | 11.5 |
| GW 451 × GW 11 (H7) | 17.2 | 9.9 | 27.1 | 9.0 | 13.6 | 11.2 | 31.2 | 10.7 |
| GW 451 × HD 2864 (H8) | 16.4 | 10.3 | 30.0 | 10.5 | 32.1 | 11.6 | 35.5 | 11.2 |
| GW 451 × UAS 385 (H9) | 15.3 | 10.0 | 26.3 | 10.6 | 25.9 | 10.6 | 34.9 | 12.0 |
| GW 496 × LOK 1 (H10) | 13.6 | 10.5 | 23.7 | 9.9 | 12.8 | 11.2 | 33.7 | 11.8 |
| GW 496 × GW 322 (H11) | 12.5 | 11.3 | 27.2 | 10.2 | 21.6 | 11.0 | 26.3 | 12.1 |
| GW 496 × GW 366 (H12) | 16.6 | 10.3 | 23.2 | 9.7 | 20.3 | 11.6 | 26.0 | 12.0 |
| GW 496 × HI 1544 (H13) | 13.2 | 10.3 | 21.9 | 10.0 | 19.3 | 11.5 | 30.0 | 10.4 |
| GW 496 × GW 173 (H14) | 12.9 | 10.2 | 41.5 | 14.1 | 13.4 | 11.2 | 26.9 | 10.8 |
| GW 496 × GW 11 (H15) | 19.1 | 10.9 | 25.2 | 9.3 | 22.2 | 11.2 | 28.3 | 9.6 |
| GW 496 × HD 2864 (H16) | 17.3 | 11.1 | 24.9 | 9.4 | 15.6 | 11.4 | 32.3 | 10.3 |
| GW 496 × UAS 385 (H17) | 16.4 | 10.9 | 27.8 | 10.7 | 16.0 | 11.2 | 26.9 | 10.9 |
| LOK 1 × GW 322 (H18) | 21.8 | 11.2 | 26.2 | 10.2 | 13.8 | 11.4 | 26.9 | 9.5 |
| LOK 1 × GW 366 (H19) | 21.3 | 9.9 | 27.2 | 10.3 | 13.2 | 11.1 | 25.9 | 10.0 |
| LOK 1 × HI 1544 (H20) | 22.1 | 11.1 | 30.5 | 10.7 | 17.0 | 11.5 | 27.2 | 9.4 |
| LOK 1 × GW 173 (H21) | 16.7 | 11.3 | 28.8 | 10.4 | 16.2 | 11.3 | 27.5 | 12.3 |
| LOK 1 × GW 11 (H22) | 13.3 | 10.9 | 25.2 | 10.3 | 10.5 | 12.4 | 28.0 | 11.6 |
| LOK 1 × HD 2864 (H23) | 12.4 | 11.5 | 25.6 | 11.0 | 23.3 | 11.6 | 26.1 | 11.1 |
| LOK 1 × UAS 385 (H24) | 14.6 | 11.0 | 25.1 | 10.0 | 35.4 | 10.6 | 26.7 | 12.3 |

 

| Table 2 (continued) | | | | | | | | |
|---|---|---|---|---|---|---|---|---|
| Entries | Normal | | | | Late | | | |
| | GYP (g) | PC (%) | Fe (ppm) | Zn (ppm) | GYP (g) | PC (%) | Fe (ppm) | Zn (ppm) |
| GW 322 × GW 366 (H25) | 18.0 | 11.3 | 26.3 | 10.3 | 13.8 | 10.7 | 26.3 | 11.6 |
| GW 322 × HI 1544 (H26) | 25.4 | 11.7 | 24.5 | 12.3 | 16.2 | 10.7 | 32.5 | 12.4 |
| GW 322 × GW 173 (H27) | 18.7 | 10.8 | 25.2 | 10.2 | 9.4 | 11.3 | 29.6 | 13.6 |
| GW 322 × GW 11 (H28) | 19.9 | 10.9 | 28.1 | 10.0 | 14.4 | 11.0 | 31.1 | 10.2 |
| GW 322 × HD 2864 (H29) | 14.5 | 11.2 | 24.5 | 10.5 | 17.5 | 11.0 | 31.6 | 11.9 |
| GW 322 × UAS 385 (H30) | 14.5 | 11.6 | 27.8 | 10.6 | 20.3 | 11.1 | 29.1 | 11.6 |
| GW 366 × HI 1544 (H31) | 16.6 | 11.3 | 25.0 | 10.5 | 12.1 | 11.4 | 32.3 | 13.0 |
| GW 366 × GW 173 (H32) | 10.7 | 10.8 | 31.8 | 11.2 | 17.1 | 10.9 | 31.8 | 12.9 |
| GW 366 × GW 11 (H33) | 13.9 | 10.9 | 23.1 | 12.2 | 17.4 | 10.9 | 26.3 | 11.4 |
| GW 366 × HD 2864 (H34) | 17.1 | 11.0 | 22.8 | 10.2 | 23.3 | 11.1 | 22.3 | 12.5 |
| GW 366 × UAS 385 (H35) | 14.2 | 10.3 | 25.2 | 9.9 | 14.9 | 11.4 | 33.4 | 12.1 |
| HI 1544 × GW 173 (H36) | 11.6 | 10.9 | 30.8 | 15.4 | 22.3 | 11.8 | 35.1 | 10.9 |
| HI 1544 × GW 11 (H37) | 12.4 | 10.0 | 25.0 | 8.8 | 17.9 | 10.8 | 36.0 | 9.7 |
| HI 1544 × HD 2864 (H38) | 15.8 | 10.5 | 28.1 | 11.2 | 17.3 | 11.1 | 23.0 | 10.7 |
| HI 1544 × UAS 385 (H39) | 15.7 | 10.2 | 25.7 | 11.1 | 14.5 | 11.0 | 28.0 | 10.9 |
| GW 173 × GW 11 (H40) | 12.7 | 10.8 | 23.3 | 11.8 | 14.2 | 11.2 | 28.7 | 12.1 |
| GW 173 × HD 2864 (H41) | 12.0 | 10.7 | 22.4 | 12.1 | 15.2 | 10.8 | 28.4 | 13.4 |
| GW 173 × UAS 385 (H42) | 11.1 | 10.7 | 25.8 | 12.4 | 10.0 | 11.0 | 29.4 | 14.3 |
| GW 11 × HD 2864 (H43) | 14.9 | 9.9 | 25.3 | 11.7 | 15.2 | 11.1 | 32.1 | 12.4 |
| GW 11 × UAS 385 (H44) | 13.7 | 10.1 | 24.9 | 10.6 | 13.9 | 10.6 | 26.9 | 12.3 |
| HD 2864 × UAS 385 (H45) | 10.0 | 10.8 | 24.4 | 11.8 | 20.2 | 10.6 | 25.9 | 14.6 |
| Mean | 16.6 | 10.7 | 26.1 | 10.6 | 17.7 | 11.2 | 29.4 | 11.5 |
| Min. | 10.0 | 9.9 | 21.7 | 8.8 | 9.4 | 10.6 | 22.3 | 9.4 |
| Max. | 32.9 | 11.7 | 41.5 | 15.4 | 35.4 | 12.4 | 36.0 | 14.6 |
| Standard checks | | | | | | | | |
| MACS 6222 –TS | 16.5 | 11.7 | 25.8 | 10.8 | 21.4 | 11.1 | 27.3 | 14.7 |
| HD 2932 –LS | 18.0 | 9.7 | 18.5 | 10.7 | 27.6 | 10.4 | 26.7 | 13.7 |
| Mean | 17.2 | 10.7 | 22.2 | 10.8 | 24.5 | 10.7 | 27.0 | 14.2 |
| Grand mean | 16.7 | 10.9 | 26.1 | 10.8 | 18.7 | 11.2 | 29.2 | 11.9 |
| S.Em. ± | 1.06 | 0.07 | 0.47 | 0.22 | 1.18 | 0.21 | 0.55 | 0.27 |
| C.D. (5%) | 2.96 | 0.18 | 1.30 | 0.61 | 3.31 | 0.59 | 1.52 | 0.74 |
| C.V.% | 10.9 | 1.03 | 3.09 | 3.51 | 10.9 | 3.27 | 3.23 | 3.84 |

**Note:**
GYP, grain yield per plant (g); PC, protein content (%); Fe, grain Fe content (ppm); Zn, grain Zn content (ppm).

conditions. Nine $F_1$ progenies for normal and late conditions exhibited significant positive SH.

## GCA effect of parents

No parent simultaneously had positive GCA effects for all the studied characters under both environments (Table 5). For GYP and PC, P1 had significant positive GCA effect

**Table 3 Estimates of heterobeltosis in individual environments for grain yield per plant, protein content, and Fe and Zn content.**

| S. N. | Hybrids | Normal | | | | Late | | | |
|---|---|---|---|---|---|---|---|---|---|
| | | Heterobeltosis | | | | | | | |
| | | GYP | PC | Fe | Zn | GYP | PC | Fe | Zn |
| 1 | GW 451 × GW 496 (H1) | 32.85** | −7.34** | −9.88** | −2.45 | −24.00** | 9.81** | 3.87 | −29.76** |
| 2 | GW 451 × LOK 1 (H2) | 37.45** | −14.53** | −22.03** | −15.01** | −24.54** | 5.25* | 4.96* | −29.60** |
| 3 | GW 451 × GW 322 (H3) | **91.15**** | −11.71** | 3.31 | −19.51** | −5.68 | 10.50** | **35.39**** | −25.12** |
| 4 | GW 451 × GW 366 (H4) | −8.59 | −11.30** | 5.79* | −23.24** | −49.01** | **10.76**** | −9.64** | −26.15** |
| 5 | GW 451 × HI 1544 (H5) | −11.51 | −15.46** | 14.50** | −31.64** | −26.65** | 8.98** | 18.08** | −26.69** |
| 6 | GW 451 × GW 173 (H6) | 52.69** | **−24.81**** | −19.06** | −24.54** | 17.41* | −2.63 | −0.88 | −22.63** |
| 7 | GW 451 × GW 11 (H7) | 3.17 | −23.44** | −13.60** | −23.76** | −32.88** | −1.52 | 8.25** | −28.01** |
| 8 | GW 451 × HD 2864 (H8) | −0.68 | −15.58** | −12.98** | −8.47** | **58.49**** | 6.11* | 7.08** | −24.89** |
| 9 | GW 451 × UAS 385 (H9) | −1.89 | −13.44** | 13.66** | −7.48** | 27.52** | −2.70 | 11.71** | −19.29** |
| 10 | GW 496 × LOK 1 (H10) | −37.27** | −13.54** | −17.47** | −14.17** | −52.27** | −0.48 | 19.43** | −15.61** |
| 11 | GW 496 × GW 322 (H11) | −27.49** | −0.79 | −5.31** | −11.82** | −11.41 | 2.01 | −0.09 | −13.76** |
| 12 | GW 496 × GW 366 (H12) | −4.00 | −8.80** | −19.35** | −12.62** | −36.77** | 10.17** | −21.78** | −14.68** |
| 13 | GW 496 × HI 1544 (H13) | −42.99** | −13.12** | −23.84** | −27.63** | −21.79** | 1.82 | 10.20** | −25.75** |
| 14 | GW 496 × GW 173 (H14) | −10.95 | −24.25** | **44.63**** | 8.87** | −36.79** | −5.21* | −14.13** | −22.93** |
| 15 | GW 496 × GW 11 (H15) | 14.67 | −15.83** | −19.56** | −21.34** | 5.35 | −1.51 | −1.79 | **−31.74**** |
| 16 | GW 496 × HD 2864 (H16) | 4.68 | −9.01** | −27.56** | −15.48** | −26.16** | 4.22 | −2.71 | −26.63** |
| 17 | GW 496 × UAS 385 (H17) | 5.53 | −2.91** | −3.33 | −3.96 | −24.41** | 2.66 | −13.99** | −22.59** |
| 18 | LOK 1 × GW 322 (H18) | 0.27 | −7.28** | −5.67** | −11.87** | −48.47** | 1.08 | −4.41 | −26.51** |
| 19 | LOK 1 × GW 366 (H19) | −1.71 | −18.44** | −2.06 | −10.46** | −59.10** | −1.00 | −22.04** | −25.24** |
| 20 | LOK 1 × HI 1544 (H20) | −4.17 | −8.14** | 9.80** | −22.90** | −36.61** | 2.35 | −3.41 | −25.72** |
| 21 | LOK 1 × GW 173 (H21) | −22.99** | −15.69** | 0.23 | −19.41** | −39.33** | −4.50 | −12.05** | −9.94** |
| 22 | LOK 1 × GW 11 (H22) | −38.60** | −15.36** | −19.50** | −12.93** | −60.74** | 9.39** | −2.71 | −15.55** |
| 23 | LOK 1 × HD 2864 (H23) | −42.89** | −5.68** | −25.54** | −4.66 | −12.99* | 2.91 | −21.34** | −18.29** |
| 24 | LOK 1 × UAS 385 (H24) | −32.64** | −9.57** | −9.57** | −13.19** | 32.46** | −5.69* | −14.48** | −9.90** |
| 25 | GW 322 × GW 366 (H25) | 4.18 | −0.11 | 11.41** | −11.09** | −57.22** | −0.71 | −20.90** | −13.31** |
| 26 | GW 322 × HI 1544 (H26) | 9.98 | −1.65** | −3.62 | −11.58** | −34.29** | −4.69 | 19.40** | −3.96 |
| 27 | GW 322 × GW 173 (H27) | 8.35 | −19.29** | −12.12** | −21.28** | −61.59** | −4.54 | −5.49** | −0.34 |
| 28 | GW 322 × GW 11 (H28) | 15.73 | −15.54** | −10.47** | −15.29** | −40.74** | −3.06 | 8.09** | −25.10** |
| 29 | GW 322 × HD 2864 (H29) | −15.60 | −8.50** | −28.73** | −8.70** | −28.04** | 1.00 | −4.87* | −12.44** |
| 30 | GW 322 × UAS 385 (H30) | −15.51 | **2.50**** | 18.12** | −8.32** | −16.70** | 1.05 | −6.89** | −15.64** |
| 31 | GW 366 × HI 1544 (H31) | −28.07** | −4.47** | −1.55 | −24.32** | **−62.50**** | 1.30 | −2.92 | −2.84 |
| 32 | GW 366 × GW 173 (H32) | −38.37** | −19.33** | 10.84** | −13.02** | −46.82** | −7.98** | −4.47* | −5.85* |
| 33 | GW 366 × GW 11 (H33) | −19.40* | −16.07** | −26.46** | 2.85 | −46.02** | −4.09 | −20.96** | −16.98** |
| 34 | GW 366 × HD 2864 (H34) | −0.89 | −10.08** | −33.85** | −6.72* | −27.69** | 2.08 | **−33.08**** | −7.67** |
| 35 | GW 366 × UAS 385 (H35) | −17.64* | −5.37** | 8.61** | −9.38** | −53.83** | 4.02 | 0.25 | −11.80** |
| 36 | HI 1544 × GW 173 (H36) | **−49.62**** | −18.57** | 7.48** | **10.67**** | −9.58 | −0.27 | 12.05** | −20.54** |
| 37 | HI 1544 × GW 11 (H37) | −46.17** | −22.67** | −20.32** | **−36.74**** | −27.58** | −5.13* | 25.08** | −29.22** |
| 38 | HI 1544 × HD 2864 (H38) | −31.45** | −14.53** | −18.40** | −19.37** | −29.86** | −1.05 | −30.59** | −21.39** |
| 39 | HI 1544 × USA 385 (H39) | −32.22** | −13.84** | 1.12 | −19.65** | −41.14** | −2.29 | −10.37** | −20.63** |

| S. N. | Hybrids | Normal | | | | Late | | | |
|---|---|---|---|---|---|---|---|---|---|
| | | Heterobeltosis | | | | | | | |
| | | GYP | PC | Fe | Zn | GYP | PC | Fe | Zn |
| 40 | GW 173 × GW 11 (H40) | −23.45** | −19.36** | −25.76** | −9.10** | −25.38** | −4.74 | −8.21** | −11.74** |
| 41 | GW 173 × HD 2864 (H41) | −27.53** | −20.22** | **−34.91**** | −6.11** | −24.29** | **−8.26**** | −14.36** | −1.88 |
| 42 | GW 173 × UAS 385 (H42) | −28.42** | −20.52** | −10.08** | −4.28 | −47.23** | −7.07** | −6.02** | 4.09 |
| 43 | GW 11 × HD 2864 (H43) | −10.50 | −23.76** | −26.65** | −0.87 | −24.56** | −1.77 | −3.30 | −9.23** |
| 44 | GW 11 × UAS 385 (H44) | −17.51* | −22.23** | −20.45** | −10.47** | −16.39 | −6.16* | −13.82** | −10.46** |
| 45 | HD 2864 × UAS 385 (H45) | −39.67** | −11.83** | −29.22** | 10.60** | 0.55 | −3.23 | −21.91** | **6.64**** |
| | Max. | 91.15 | 2.50 | 44.63 | 10.67 | 58.49 | 10.76 | 35.39 | 6.64 |
| | Min. | −49.62 | −24.81 | −34.91 | −36.74 | −62.50 | −8.26 | −33.08 | −31.74 |
| | S.E. ± | 1.43 | 0.08 | 0.56 | 0.30 | 1.57 | 0.30 | 0.65 | 0.32 |
| Number of significant crosses | | 24 | 43 | 38 | 38 | 39 | 15 | 34 | 39 |
| Significant positive crosses | | 4 | 1 | 10 | 3 | 4 | 8 | 12 | 1 |
| Significant negative crosses | | 20 | 42 | 28 | 35 | 35 | 7 | 22 | 38 |

**Note:**
*, ** Significant at 5% and 1% levels, respectively; highest and lowest values in bold; HB, heterobeltosis; $SH_1$, standard heterosis over check variety MACS 6222 (TS); $SH_2$, standard heterosis over check variety HD 2932 (LS).

under both environments, whereas P7 had the highest and most significant positive effects for all environments except GYP in $E_1$. Hence, this parent is considered a good general component. P10 and P4 were considered poor general components because they possessed a significant negative GCA effect under both environments (Table 5). For Zn content, the numbers of parents depicted significant and positive GCA effects were two in $E_1$ and four in $E_2$. GW 173 with a negative GCA effect for grain yield had positive GCA effects for micronutrients under both environments. The GCA effects of few parents was not in same direction (negative or positive) across the environments for all traits (HD 2864 for GYP; GW 451, GW 322, for PC; GW 451, GW 496, GW 366, HI 1544, UAS 385 for Zn).

## SCA effect of hybrids

As shown in Table 6, the hybrid crosses did not simultaneously obtain positive SCA effects for all studied characters. For Fe and Zn content, SCA effects for hybrids were consistently positive under both environments, although every hybrid had a different reaction. For GYP, the hybrid combinations H15 and H34 exhibited positive significant SCA effects under both environments. The hybrids (H3 > H2 > H6 > H26) exhibited significant ($P < 0.01$) positive SCA effects for GYP under E1.

Moreover, H3 had the greatest positive SCA effect for GYP (10.73**) under $E_1$ and significant & positive (2.71**) under $E_2$; conversely, for PC (0.69**) and Fe content (3.78**), there was a positive SCA effect under $E_2$. A few hybrid combinations with a negative SCA effect under $E_1$ had a positive SCA effect under $E_2$. For example, the hybrid combinations H2, H16, H17, H19, and H26 for GYP, H24, H25, and H26 for PC; H14, H17, H19, and H20 for Fe content, and H1, H9, and H14 for Zn exhibited a positive SCA effect under $E_1$, thus, these crosses were considered good specific components. When

**Table 4 Estimates of standard heterosis in individual environments for grain yield per plant, protein content, and Fe and Zn content.**

| Sr. No. | Hybrids | Normal | | | | | | | | Late | | | | | | | |
|---|---|---|---|---|---|---|---|---|---|---|---|---|---|---|---|---|---|
| | | Standard heterosis 1 | | | | Standard heterosis 2 | | | | Standard heterosis 1 | | | | Standard heterosis 2 | | | |
| | | GYP | PC | Fe | Zn | GYP | PC | Fe | Zn | GYP | PC | Fe | Zn | GYP | PC | Fe | Zn |
| 1 | GW 451 × GW 496 (H1) | 20.94* | −8.95** | 0.34 | 3.43 | 10.86 | 9.82** | 39.93** | 4.39 | −25.00** | 3.59 | 0.00 | −28.98** | −41.85** | 10.56** | 2.25 | −23.80** |
| 2 | GW 451 × LOK 1 (H2) | 80.87** | −11.43** | −16.04** | −9.44** | 65.80** | 6.83** | 17.10** | −8.60** | −5.64 | 6.76* | 8.33** | −28.82** | −26.84** | 13.95** | 10.76** | −23.63** |
| 3 | GW 451 × GW 322 (H3) | 99.41** | −13.24** | −5.60** | −14.04** | 82.79** | 4.65* | 31.65** | −13.24** | 7.39 | 7.35* | 22.84** | −24.29** | −16.73** | 14.58** | 25.61** | −18.76** |
| 4 | GW 451 × GW 366 (H4) | −4.20 | −12.84** | −6.05** | −18.62** | −12.18 | 5.13* | 31.02** | −17.86** | −23.33** | 4.94 | 10.11** | −25.33** | −40.56** | 12.01** | 12.58** | −19.88** |
| 5 | GW 451 × HI 1544 (H5) | 23.89** | −14.38** | 12.76** | −12.19** | 13.57 | 3.28* | 57.26** | −11.37** | −15.30* | 10.60** | 17.86** | −25.87** | −34.32** | 18.04** | 20.51** | −20.46** |
| 6 | GW 451 × GW 173 (H6) | 39.00** | −13.78** | −9.97** | −9.66** | 27.42** | 3.99* | 25.55** | −8.82** | 11.25 | 3.45 | 13.57** | −21.77** | −13.74* | 10.42** | 16.12** | −16.06** |
| 7 | GW 451 × GW 11 (H7) | 4.07 | −15.36** | 4.99* | −16.60** | −4.60 | 2.10* | 46.42** | −15.83** | −36.40** | 0.68 | 14.25** | −27.21** | −50.69** | 7.46** | 16.82** | −21.90** |
| 8 | GW 451 × HD 2864 (H8) | −0.46 | −11.74** | 16.15** | −2.96 | −8.76 | 6.45** | 61.98** | −2.06 | 50.17** | 4.38 | 30.12** | −24.06** | 16.44** | 11.41** | 33.05** | −18.52** |
| 9 | GW 451 × UAS 385 (H9) | −7.54 | −14.94** | 2.13 | −1.91 | −15.24 | 2.60* | 42.43** | −1.00 | 20.83** | −4.08 | 27.89** | −18.39** | −6.32 | 2.38 | 30.76** | −12.43** |
| 10 | GW 496 × LOK 1 (H10) | −17.45* | −10.40** | −8.11** | −8.56** | −24.33** | 8.07** | 28.15** | −7.70** | −40.31** | 0.95 | 23.26** | −19.46** | −53.72** | 7.74** | 26.03** | −13.58** |
| 11 | GW 496 × GW 322 (H11) | −24.36** | −3.75* | 5.43* | −5.83* | −30.66** | 16.10** | 47.04** | −4.95 | 0.87 | −0.90 | −3.81 | −17.69** | −21.79** | 5.77* | −1.65 | −11.68** |
| 12 | GW 496 × GW 366 (H12) | 0.62 | −12.37** | −10.21** | −10.06** | −7.76 | 5.70* | 25.22** | −9.22** | −4.92 | 4.38 | −4.69* | −18.57** | −26.28** | 11.41** | −2.55 | −12.63** |
| 13 | GW 496 × HI 1544 (H13) | −20.18* | −12.01** | −15.21** | −7.04** | −26.84** | 6.14* | 18.25** | −6.17* | −9.69 | 3.33 | 10.00** | −29.14** | −29.98** | 10.28** | 12.47** | −23.97** |
| 14 | GW 496 × GW 173 (H14) | −21.88* | −13.15** | 61.03** | 30.34** | −28.39** | 4.76** | 124.58** | 31.56** | −37.62** | 0.72 | −1.61 | −26.44** | −51.63** | 7.50** | 0.60 | −21.07** |
| 15 | GW 496 × GW 11 (H15) | 15.67 | −6.93** | −2.24 | −13.96** | 6.03 | 12.25** | 36.33** | −13.15** | 3.97 | 0.68 | 3.65 | −34.85** | −19.39** | 7.46** | 5.98* | −30.10** |
| 16 | GW 496 × HD 2864 (H16) | 4.91 | −4.88* | −3.31 | −13.00** | −3.83 | 14.74** | 34.84** | −12.19** | −27.13** | 2.53 | 18.22** | −29.98** | −43.50** | 9.44** | 20.87** | −24.87** |
| 17 | GW 496 × UAS 385 (H17) | −0.55 | −6.71** | 7.64** | −1.14 | −8.83 | 12.52** | 50.11** | −0.22 | −25.40** | 1.21 | −1.54 | −26.12** | −42.16** | 8.03** | 0.67 | −20.73** |
| 18 | LOK 1 × GW 322 (H18) | 31.95** | −3.91* | 1.59 | −5.89* | 20.95** | 15.90** | 41.67** | −5.01 | −35.56** | 2.53 | −1.34 | −35.56** | −50.04** | 9.43** | 0.87 | −30.85** |
| 19 | LOK 1 × GW 366 (H19) | 29.34** | −15.48** | 5.47* | −4.60 | 18.56* | 1.95* | 47.09** | −3.71 | −38.50** | 0.42 | −5.01* | −31.84** | −52.31** | 7.18* | −2.87 | −26.86** |
| 20 | LOK 1 × HI 1544 (H20) | 34.17** | −4.80** | 18.24** | −0.96 | 22.99** | 14.82** | 64.90** | −0.03 | −20.73** | 3.87 | −0.31 | −35.94** | −38.54** | 10.86** | 1.94 | −31.27** |
| 21 | LOK 1 × GW 173 (H21) | 1.33 | −3.32* | 11.49** | −3.52 | −7.11 | 16.61** | 55.48** | −2.62 | −24.13** | 1.47 | 0.77 | −16.19** | −41.18** | 8.30** | 3.03 | −10.07** |
| 22 | LOK 1 × GW 11 (H22) | −19.20* | −6.42** | −2.17 | −4.76 | −25.94** | 12.88** | 36.43** | −3.87 | −50.91** | 11.83** | 2.69 | −21.41** | −61.93** | 19.35** | 4.99* | −15.67** |
| 23 | LOK 1 × HD 2864 (H23) | −24.85** | −1.40* | −0.61 | 1.57 | −31.11** | 18.93** | 38.61** | 2.52 | 8.80 | 4.38 | −4.42 | −24.65** | −15.64** | 11.41** | −2.27 | −19.15** |
| 24 | LOK 1 × UAS 385 (H24) | −11.36 | −6.29** | −2.62 | −7.51 | −18.75* | 13.03** | 35.81** | −6.65 | 65.64** | −4.34 | −2.10 | −16.01** | 28.43** | 2.10 | 0.10 | −9.88** |
| 25 | GW 322 × GW 366 (H25) | 9.19 | −3.09** | 1.80 | −5.06 | 0.09 | 16.89** | 41.98** | −4.17 | −35.67** | −3.54 | −3.61 | −20.95** | −50.12** | 2.95 | −1.45 | −15.18** |
| 26 | GW 322 × HI 1544 (H26) | 53.99** | −0.39 | −5.08* | 13.58** | 41.16** | 20.15** | 32.37** | 14.64** | −24.13** | −3.28 | 19.18** | −15.78** | −41.17** | 3.23 | 21.86** | −9.64** |
| 27 | GW 322 × GW 173 (H27) | 13.03 | −7.45** | −2.26 | −5.76* | 3.61 | 11.63** | 36.31** | −4.88 | −56.26** | 1.42 | 8.29** | −7.26** | −66.09** | 8.25** | 10.72** | −0.49 |
| 28 | GW 322 × GW 11 (H28) | 20.73* | −6.62** | 8.80** | −7.35** | 10.67 | 12.63** | 51.73** | −6.48* | −32.52** | −0.90 | 14.08** | −30.29** | −47.68** | 5.77* | 16.64** | −25.21** |
| 29 | GW 322 × HD 2864 (H29) | −11.95 | −4.34* | −4.87* | −2.50 | −19.29* | 15.38** | 32.67** | −1.59 | −18.07* | −0.64 | 15.59** | −19.25** | −36.47** | 6.05* | 18.19** | −13.36** |
| 30 | GW 322 × UAS 385 (H30) | −11.86 | −0.56 | 7.93** | −2.10 | −19.20* | 19.95** | 50.52** | −1.18 | −5.16 | −0.37 | 6.59** | −21.36** | −26.46** | 6.33* | 8.99** | −15.62** |
| 31 | GW 366 × HI 1544 (H31) | 0.71 | −3.24* | −3.04 | −2.78 | −7.68 | 16.71** | 35.22** | −1.87 | −43.61** | 2.80 | 18.30** | −11.41** | −56.28** | 9.72** | 20.96** | −4.94* |
| 32 | GW 366 × GW 173 (H32) | −35.41** | −7.50** | 23.28** | 4.14 | −40.79** | 11.57** | 71.93** | 5.11 | −20.03** | −2.22 | 16.41** | −12.38** | −38.00** | 4.36 | 19.03** | −5.99** |
| 33 | GW 366 × GW 11 (H33) | −15.53 | −7.20** | −10.63** | 12.50** | −22.57** | 11.93** | 24.64** | 13.55** | −18.83* | −1.96 | −3.69 | −22.74** | −37.07** | 4.64 | −1.52 | −17.10** |
| 34 | GW 366 × HD 2864 (H34) | 3.87 | −6.00** | −11.71** | −5.68* | −4.78 | 13.38** | 23.13** | −4.80 | 8.73 | 0.42 | −18.45** | −14.85** | −15.70* | 7.18** | −16.62** | −8.64** |
| 35 | GW 366 × UAS 385 (H35) | −13.68 | −11.91** | −2.40* | −8.37** | −20.87** | 6.25* | 36.11** | −7.52** | −30.58** | 2.55 | 22.16** | −17.78** | −46.17** | 9.46** | 24.91** | −11.78** |
| 36 | HI 1544 × GW 173 (H36) | −29.46** | −6.63* | 19.55** | 42.16** | −35.34** | 12.62** | 66.72** | 43.49** | 4.41 | 5.97* | 28.39** | −26.05** | −19.04** | 13.10** | 31.27** | −20.66** |
| 37 | HI 1544 × GW 11 (H37) | −24.63** | −14.50** | −3.17 | −18.73** | −30.91** | 3.13* | 35.04** | −17.98** | −16.37* | −3.02 | 32.01** | −34.13** | −35.16** | 3.51 | 34.98** | −29.32** |
| 38 | HI 1544 × HD 2864 (H38) | −4.02 | −10.65** | 8.91** | 3.58 | −12.02 | 7.77** | 51.89** | 4.55 | −19.00** | 0.42 | −15.65** | −27.51** | −37.20** | 7.18* | −13.76** | −22.21** |

| Sr. No. | Hybrids | Normal | | | | | | | | Late | | | | | | | |
| --- | --- | --- | --- | --- | --- | --- | --- | --- | --- | --- | --- | --- | --- | --- | --- | --- | --- |
| | | Standard heterosis 1 | | | | Standard heterosis 2 | | | | Standard heterosis 1 | | | | Standard heterosis 2 | | | |
| | | GYP | PC | Fe | Zn | GYP | PC | Fe | Zn | GYP | PC | Fe | Zn | GYP | PC | Fe | Zn |
| 39 | HI 1544 × UAS 385 (H39) | −5.10 | −12.74** | −0.42 | 3.21 | −13.01 | 5.26** | 38.87** | 4.17 | −32.03** | −0.84 | 2.61 | −26.01** | −47.30** | 5.83* | 4.92* | −20.61** |
| 40 | GW 173 × GW 11 (H40) | −22.78** | −7.53** | −9.78** | 8.83** | −29.21** | 11.53** | 25.82** | 9.84** | −33.85** | 1.21 | 5.18* | −17.87** | −48.71** | 8.03** | 7.54** | −11.87** |
| 41 | GW 173 × HD 2864 (H41) | −27.38** | −8.52** | −13.12** | 12.41** | −33.43** | 10.34** | 21.17** | 13.46** | −28.89** | −2.52 | 4.07 | −8.68** | −44.87** | 4.04 | 6.40** | −2.02 |
| 42 | GW 173 × UAS 385 (H42) | −32.54** | −8.87** | 0.02 | 14.60** | −38.16** | 9.92** | 39.49** | 15.67** | −53.22** | −1.26 | 7.68** | −2.97 | −63.73** | 5.39 | 10.10** | 4.11 |
| 43 | GW 11 × HD 2864 (H43) | −9.72 | −15.71** | −2.10 | 8.43** | −17.24* | 1.67* | 36.53** | 9.44** | −29.14** | 0.42 | 17.51** | −15.53** | −45.06** | 7.18* | 20.15** | −9.37** |
| 44 | GW 11 × UAS 385 (H44) | −16.79 | −14.01** | −3.32 | −2.07 | −23.72** | 3.72** | 34.82** | −1.15 | −34.92** | −4.07 | −1.34 | −16.53** | −49.54** | 2.38 | 0.87 | −10.44** |
| 45 | HD 2864 × UAS 385(H45) | −39.54** | −7.82** | −5.52* | 9.14** | −44.58** | 11.18** | 31.76** | 10.16** | −5.56 | −4.60 | −5.10* | −0.59 | −26.78** | 1.82 | −2.97 | 6.67** |
| | Max. | 99.41 | −0.39 | 61.03 | 42.16 | 82.79 | 20.15 | 124.58 | 43.49 | 65.64 | 11.83 | 32.01 | −0.59 | 28.43 | 19.35 | 34.98 | 6.67 |
| | Min. | −39.54 | −15.71 | −16.04 | −18.73 | −44.58 | 1.67 | 17.10 | −17.98 | −56.26 | −4.60 | −18.45 | −35.94 | −66.09 | 1.82 | −16.62 | −31.27 |
| | S.E. ± | 1.43 | 0.08 | 0.56 | 0.30 | 1.43 | 0.08 | 0.56 | 0.30 | 1.57 | 0.30 | 0.65 | 0.32 | 1.57 | 0.30 | 0.65 | 0.32 |
| | Number of significant crosses | 22 | 43 | 29 | 27 | 28 | 45 | 45 | 23 | 32 | 5 | 29 | 42 | 44 | 34 | 30 | 42 |
| | Significant positive crosses | 10 | 0 | 14 | 9 | 7 | 45 | 45 | 9 | 3 | 5 | 24 | 0 | 2 | 34 | 28 | 1 |
| | Significant negative crosses | 12 | 43 | 15 | 18 | 21 | 0 | 0 | 14 | 29 | 0 | 5 | 43 | 42 | 0 | 2 | 41 |

**Note:**
*, ** Significant at 5% and 1% levels, respectively; highest and lowest values in bold; HB, heterobeltosis; SH₁, standard heterosis over check variety MACS 6222 (TS); SH₂, standard heterosis over check variety HD 2932 (LS).

**Table 5 Estimates of general combining ability (GCA) effects of parents for grain yield per plant, protein content, and Fe and Zn content.**

| Characters | | GYP | | PC | | IC | | ZC | |
|---|---|---|---|---|---|---|---|---|---|
| S. N. | Parents | $E_1$ | $E_2$ | $E_1$ | $E_2$ | $E_1$ | $E_2$ | $E_1$ | $E_2$ |
| 1 | GW 451 (P1) | **3.06**** | **2.17**** | **−0.42**** | 0.16** | −1.05** | **1.03**** | **−0.63**** | −0.12 |
| 2 | GW 496 (P2) | −1.17** | −0.35 | −0.10** | −0.04 | 0.83** | −1.03** | −0.18* | −0.34** |
| 3 | LOK 1 (P3) | 2.10** | 1.02** | 0.20** | **0.20**** | 0.07 | **−1.20**** | −0.28** | −0.55** |
| 4 | GW 322 (P4) | 2.39** | −0.41 | 0.27** | −0.12* | −0.62** | −0.59** | −0.20** | −0.05 |
| 5 | GW 366 (P5) | −0.42 | 0.75* | −0.16** | −0.11 | **−1.19**** | −0.13 | −0.33** | 0.23** |
| 6 | HI 1544 (P6) | 1.29** | 0.05 | 0.01 | 0.13* | 0.21 | 0.70** | 0.70** | **−0.61**** |
| 7 | GW 173 (P7) | −2.14** | −2.01** | **0.29**** | 0.11 | **1.79**** | 0.70** | **1.19**** | **0.66**** |
| 8 | GW 11 (P8) | −1.12** | **−2.59**** | 0.04* | 0.01 | 0.08 | 0.33* | −0.14* | −0.27** |
| 9 | HD 2864 (P9) | −1.70** | 1.35** | 0.14** | −0.07 | 0.67** | 0.09 | 0.07 | 0.39** |
| 10 | UAS 385 (P10) | **−2.28**** | 0.02 | −0.25** | **−0.26**** | −0.78** | 0.11 | −0.21** | 0.65** |
| Range | Min. | −2.28 | −2.59 | −0.42 | −0.26 | −1.19 | −1.20 | −0.63 | −0.61 |
| | Max. | 3.06 | 2.17 | 0.29 | 0.20 | 1.79 | 1.03 | 1.19 | 0.66 |
| S.E. (gi) ± | | 0.29 | 0.32 | 0.02 | 0.06 | 0.12 | 0.15 | 0.06 | 0.07 |
| Number of significant parents | | 9 | 6 | 9 | 5 | 7 | 7 | 9 | 8 |
| Significant and positive parents | | 4 | 4 | 5 | 3 | 3 | 4 | 2 | 4 |
| Significant and negative parents | | 5 | 2 | 4 | 2 | 4 | 3 | 7 | 4 |

**Note:**
*, ** Significant at 5% and 1% levels, respectively; highest and lowest values are in bold.

negative effects under $E_2$, therefore, they were considered poor specific components. For Fe and Zn content, H43 had a significant positive SCA effect under both environments but in reverse order for GYP and PC. H14 had the highest positive significant results for Fe and Zn content under $E_1$. For Fe content, 16 hybrids under $E_1$ and $E_2$ each exhibited significant positive SCA effects. For Zn content, eight of 45 hybrids exhibited significant positive SCA effects under $E_1$ and $E_2$. For Zn content, H36 had the highest significant positive SCA effects under $E_1$ (2.66**), and significant positive effects in both environments for Fe content. In addition, the variance caused by SCA ($\sigma^2_{SCA}$) was greater than GCA ($\sigma^2_{GCA}$) for all studied environments (Table 1).

## Correlations among *per se* performance, heterosis, and combining ability

The interconnection among *per se* performance, heterosis, and combining ability effects yield attributes, PC, and Fe and Zn content under both environments are shown in Table S3. SCA had no significant correlation with GCA effects, except for Zn content under the late condition. BPH had a significant relationship with GYP (r = 0.41) under normal conditions. When Fe and Zn contents were significantly correlated under the late condition. For SH1, SH2, and phenotype, significant relationships were observed for all traits under both environments, except Fe under the normal condition. SCA had a positive significant ($P < 0.001$) correlation with BPH, SH1, SH2, and phenotype for yield component traits and grain protein, Fe, and Zn concentrations under both environments. GSCA had a significantly ($P < 0.01$) positive association with the GCA effect. On the
Table 6 Estimates of specific combining ability (SCA) effect of $F_1$'s for grain yield per plant, protein content, and Fe and Zn content.

| Characters | | GYP | | PC | | IC | | ZC | |
|---|---|---|---|---|---|---|---|---|---|
| S. N. | Hybrids | $E_1$ | $E_2$ | $E_1$ | $E_2$ | $E_1$ | $E_2$ | $E_1$ | $E_2$ |
| 1 | GW 451 × GW 496 (H1) | 1.33 | −4.29** | 0.29** | 0.19 | −0.15 | −2.02** | 1.18** | −0.96** |
| 2 | GW 451 × LOK 1 (H2) | 7.95** | −1.51 | −0.30** | 0.30 | −3.61** | 0.43 | −0.11 | −0.73** |
| 3 | GW 451 × GW 322 (H3) | **10.73**** | 2.71* | −0.58** | 0.69** | −0.23 | 3.78** | −0.69** | −0.55* |
| 4 | GW 451 × GW 366 (H4) | −3.56** | −5.03** | −0.10* | 0.41* | 0.22 | −0.15 | −1.05** | −1.00** |
| 5 | GW 451 × HI 1544 (H5) | −0.65 | −2.61* | −0.44** | 0.80** | 3.67** | 1.13* | −1.38** | −0.23 |
| 6 | GW 451 × GW 173 (H6) | 5.29** | 5.13** | −0.66** | 0.03 | −3.77** | −0.04 | −1.61** | −0.90 |
| 7 | GW 451 × GW 11 (H7) | −1.50 | −4.49** | −0.59** | −0.17 | 1.80** | 0.52 | −1.03** | −0.77** |
| 8 | GW 451 × HD 2864 (H8) | −1.66 | 10.10** | −0.28** | 0.31 | 4.09** | 5.09** | 0.24 | −0.97** |
| 9 | GW 451 × UAS 385 (H9) | −2.26* | 5.15** | −0.26** | −0.44* | 1.93** | 4.46** | 0.63** | −0.39 |
| 10 | GW 496 × LOK 1 (H10) | −4.04** | −6.41** | −0.50** | −0.15 | −3.45** | **6.56**** | −0.47* | 0.87** |
| 11 | GW 496 × GW 322 (H11) | **−5.46**** | 3.83** | 0.21** | −0.03 | 0.74 | −1.44** | −0.26 | 0.64** |
| 12 | GW 496 × GW 366 (H12) | 1.47 | 1.43 | −0.37** | 0.54** | −2.73** | −2.13** | −0.58** | 0.22 |
| 13 | GW 496 × HI 1544 (H13) | −3.68** | 1.11 | −0.49** | 0.19 | −5.42** | 1.04** | −1.28** | −0.49* |
| 14 | GW 496 × GW 173 (H14) | −0.53 | −2.80* | −0.91** | −0.08 | **12.67**** | −2.13** | 2.26** | −1.37** |
| 15 | GW 496 × GW 11 (H15) | 4.65** | 6.67** | 0.07 | 0.02 | −1.95** | −0.32 | −1.19** | −1.68** |
| 16 | GW 496 × HD 2864 (H16) | 3.46** | −3.93** | 0.20** | 0.29 | −2.81** | 3.90** | −1.30** | −1.62** |
| 17 | GW 496 × UAS 385 (H17) | 3.13** | −2.22* | 0.38** | 0.33 | 1.47** | −1.51** | 0.26 | −1.31** |
| 18 | LOK 1 × GW 322 (H18) | 0.56 | −5.33** | −0.11* | 0.11 | 0.51 | −0.59 | −0.17 | **−1.78**** |
| 19 | LOK 1 × GW 366 (H19) | 2.93** | −7.12** | −1.03** | −0.14 | 2.08** | −2.04** | 0.11 | −1.52** |
| 20 | LOK 1 × HI 1544 (H20) | 2.02* | −2.62* | 0.06 | 0.01 | 3.97** | −1.60** | −0.53** | −1.28** |
| 21 | LOK 1 × GW 173 (H21) | 0.03 | −1.28 | −0.06 | −0.24 | 0.65 | −1.30* | −1.30** | 0.35 |
| 22 | LOK 1 × GW 11 (H22) | −4.38** | −6.44** | −0.17** | **1.02**** | −1.17** | −0.40 | −0.10 | 0.51* |
| 23 | LOK 1 × HD 2864 (H23) | −4.72** | 2.40* | 0.31** | 0.27 | −1.35** | −2.10** | 0.37 | −0.63** |
| 24 | LOK 1 × UAS 385 (H24) | −1.92* | **15.89**** | 0.13** | −0.52** | −0.42 | −1.49** | −0.33 | 0.38 |
| 25 | GW 322 × GW 366 (H25) | −0.68 | −5.09** | 0.35** | −0.25 | 1.82** | −2.27** | −0.02 | −0.42 |
| 26 | GW 322 × HI 1544 (H26) | 5.00** | −1.91 | 0.51** | −0.46** | −1.36** | 3.11** | 0.96** | 1.19** |
| 27 | GW 322 × GW 173 (H27) | 1.68 | −6.73** | −0.61** | 0.08 | −2.21** | 0.14 | −1.62** | 1.17** |
| 28 | GW 322 × GW 11 (H28) | 1.93* | −1.07 | −0.26** | −0.07 | 2.35** | 2.10** | −0.46* | −1.29** |
| 29 | GW 322 × HD 2864 (H29) | −2.88** | −1.92 | −0.10* | 0.03 | −1.76** | 2.75** | −0.15 | −0.33 |
| 30 | GW 322 × UAS 385 (H30) | −2.29* | 2.17* | **0.73**** | 0.25 | 3.00** | 0.27 | 0.17 | −0.90** |
| 31 | GW 366 × HI 1544 (H31) | −0.98 | **−7.25**** | 0.60** | 0.20 | −0.27 | 2.41** | −0.67** | 1.54** |
| 32 | GW 366 × GW 173 (H32) | −3.51** | −0.14 | −0.18** | −0.34 | 4.95** | 1.90** | −0.42* | 0.13 |
| 33 | GW 366 × GW 11 (H33) | −1.25 | 0.69 | 0.10* | −0.20 | −2.09** | −3.21** | 1.81** | −0.47* |
| 34 | GW 366 × HD 2864 (H34) | 2.54** | 2.64* | 0.13** | 0.13 | −2.96** | −7.00** | −0.36 | 0.03 |
| 35 | GW 366 × UAS 385 (H35) | 0.22 | −4.44** | −0.17** | 0.56** | 0.89* | 4.06** | −0.37 | −0.66** |
| 36 | HI 1544 × GW 173 (H36) | −4.24** | 5.79** | −0.25** | 0.33 | 2.58** | 4.33** | **2.66** | −1.04** |
| 37 | HI 1544 × GW 11 (H37) | −4.46** | 1.92 | −0.92** | **−0.56**** | −1.57** | 5.70** | **−2.59** | −1.30** |
| 38 | HI 1544 × HD 2864 (H38) | −0.48 | −2.59* | −0.57** | −0.10 | 0.96* | **−7.08**** | −0.39 | −0.98** |
| 39 | HI 1544 × UAS 385 (H39) | −0.08 | −4.04** | −0.43** | −0.06 | 0.00 | −2.11** | −0.15 | −1.03** |
| 40 | GW 173 × GW 11 (H40) | −0.72 | 0.24 | −0.39** | −0.07 | −4.85** | −1.63** | −0.11 | −0.18 |

(Continued)

**Table 6** (continued)

| Characters | | GYP | | PC | | IC | | ZC | |
|---|---|---|---|---|---|---|---|---|---|
| S. N. | Hybrids | $E_1$ | $E_2$ | $E_1$ | $E_2$ | $E_1$ | $E_2$ | $E_1$ | $E_2$ |
| 41 | GW 173 × HD 2864 (H41) | −0.90 | −2.64* | −0.61** | −0.41* | **−6.30**** | −1.69** | 0.07 | 0.51* |
| 42 | GW 173 × UAS 385 (H42) | −1.18 | −6.52** | −0.26** | −0.08 | −1.46** | −0.72 | 0.59** | 1.09** |
| 43 | GW 11 × HD 2864 (H43) | 1.00 | −2.12 | **−1.20**** | 0.02 | 1.75** | 2.36** | 0.97** | 0.43 |
| 44 | GW 11 × UAS 385 (H44) | 0.40 | −2.03 | −0.60** | −0.29 | −0.62 | −2.81** | 0.12 | 0.03 |
| 45 | HD 2864 × UAS 385 (H45) | −2.77** | 0.31 | 0.00 | −0.28 | −1.77** | −3.60** | 1.12** | **1.71**** |
| Range | Min. | −5.46 | −7.25 | −1.20 | −0.56 | −6.30 | −7.08 | −2.59 | −1.78 |
| | Max. | 10.73 | 15.89 | 0.73 | 1.02 | 12.67 | 6.56 | 2.66 | 1.71 |
| S.E. (sij) ± | | 0.96 | 1.09 | 0.05 | 0.20 | 0.41 | 0.51 | 0.20 | 0.22 |
| Number of significant crosses | | 25 | 31 | 41 | 11 | 35 | 35 | 27 | 32 |
| Significant and positive crosses | | 11 | 11 | 12 | 6 | 16 | 16 | 8 | 8 |
| Significant and negative crosses | | 14 | 20 | 29 | 5 | 19 | 19 | 19 | 24 |

**Note:**
*, ** Significant at 5% and 1% levels, respectively; highest and lowest values are in bold.

contrary, it had a negative interaction with SCA, BPH, SH1, and SH2. GSCA of BPH was positively associated, except for PC and Zn content under the normal condition. Under both conditions, the GSCA had positively significant interactions with SH1, SH2, and phenotype for all traits, except Fe under the normal condition.

## DISCUSSION

Approximately two billion people, nearly one third of the global population, have been estimated to be deficient in one or more mineral components (*Huseynova & Rustamova, 2010*). Biofortification is one of the most effective approaches to alleviate malnutrition (Global Nutrition Report, 2017, https://www.globalnutritionreport.org). Besides the major required elements, micronutrients are also involved in regulating many vital metabolic functions (*Heck et al., 2020*). Biofortification is an approved strategy to fight micronutrient deficiency in large populations, particularly for those living in developing countries.
To make it more effectual, accessible, and manageable for people, well planned, applied, observed, and appraised biofortification programs are needed to produce economical and socially allowable biofortified food crops (*Bouis & Saltzman, 2017*; *Bouis & Welch, 2010*; *Bouis, Saltzman & Birol, 2019*).

Significant advancement has been attained in developing high Fe and Zn wheat varieties for cultivation under normal and late conditions. Biofortified varieties, such as HD 3298, WB 02, HPBW 01, Pusa Tejas (HI 8759) Pusa Ujala (HI 1605) HD 3171, HI 8777, MACS 4028, PBW 752, PBW 757, Karan Vandana (DBW 187), DBW 173, UAS 375, DDW 47, PBW 771, HI 8802, HI 8805, HD 3249, MACS 4058, HI 1633, DBW 303, and DDW 48 with high PC, Fe, and Zn content have been released for commercial farming in different countries (*Yadava et al., 2020*). In wheat, various studies on PC, Fe, and Zn discuss biofortification. Some progress in grain yield advancement and micronutrient improvement has been made under both environments discussed in the present study.

In this study, a total of 45 hybrids were developed by following the half diallel mating design by using 10 parental genotypes under normal and late environments. The hybrids, parental genotypes and two standard checks, MACS 6222—TS and HD 2932—LS, were estimated for GYP and protein, Fe, and Zn content under normal and late sown conditions. The parents, P9 and P5, were good general combiners for all the studied characters under both environments. For Fe and Zn traits, hybrid H14 was best under normal sowing and H37 for late sowing. In our study, some crosses were better for GYP than their parents and check varieties, and expressed the distinct particular cross yield under both environments. The hybrids exhibited a smaller mean value for protein content compared to the parents. In the present study, several hybrids manifested noticeable heterotic response over the better parent (heterobeltosis) and standard check varieties (standard heterosis). For example, MACS 6222 was best for normal sown and HD 2932 for late sown. Between the two groups of heterosis, SH was more appropriate than BPH. SH illustrated the acceptance yield edge of hybrid varieties over the best checks. Under both environments, hybrids exhibited negative heterosis for all the calculated characters.

For PC, the extent of HB was low to high in the negative direction and low in the positive direction. However, the magnitude of SH was moderate in the positive and low to high in the negative direction. The findings agreed with the results of previous studies (*Kumar, 2012*; *Desale & Mehta, 2013*; *Chaudhari & Patel, 2014a*; *Ekhlaque, Kamar & Jaiswal, 2016*) for HB. The findings for SH were in accordance with those of *Kumar et al. (2018)* and *Chaudhari et al. (2017)*. For Fe content, the magnitude of HB was moderate in the negative and low in the positive direction. The magnitude of SH was moderate in both directions. The results contradicted the finding of *Younas et al. (2019)*. For Zn content, the magnitude of HB and SH was high in the negative and low in positive direction.

The differential parental lines using GCA illustrated the average performance of the parental line in an array of hybrid crosses. In this study, the effects of GCA in parental lines for protein content varied with the environment. The P1 had the highest significant positive GCA effect in $E_2$ (0.20). Another good general component was P7 in $E_1$ (0.29). P10 was considered a poor general component because it possessed a significant negative GCA effect under all environments. Under both environments, the effects of GCA were at odds with the grain Fe content of parents, excluding P7. Under $E_1$, the effects of GCA for Fe and Zn contents were contradictory. Under $E_1$ and $E_2$, grain yield and Fe and Zn content showed an anti-positive correlation and GCA effect such that these traits can be considered to improve parental substances. Hence, these types of parents could be used in selection of desirable types to improve combination breeding.

This study revealed a high degree of correlation between the performance of parents and their GCA effects for most of the characters studied, and whether the desirable *per se* performance could be used for the hybridization program to select parents.

The success in crop improvement lies in isolating superior gene combinations in the genotypes with high combing ability. The genotypes with good general combining ability can be further exploited in developing a new variety.

Non-additive gene action is linked with SCA, followed by dominance over dominance and epistatic effects (*Younas et al., 2019*; *Sprague & Tatum, 1942*; *Saleem, Mirza & Haq,*

2010; *Chaudhari et al., 2022*). The findings also indicated the involvement of both additive and non-additive gene action in the inheritance of yield and related traits in wheat (*Chaudhari et al., 2023*). Analysis of the crosses revealed that no special combination prevailed in significant positive SCA values for all the estimated characters concurrently. This result is in accordance with those from previous studies on rice (*Griffing, 1956*; *Gramaje et al., 2020*), maize (*Tiwari et al., 2011*), and bread wheat (*Kahriman et al., 2016*). However, the crosses H3, H2, and H6 (significant ($P < 0.01$) for GYP) and H43 (significant ($P < 0.01$) for grain Fe and Zn content) showed positive SCA values for all the traits in normal conditions. Oppositely, late condition-H3 (significant for GYP, PC, and grain Fe content) and H26 for grain Fe and Zn content appeared to have positive SCA. These results were in accordance with previous studies on rice (*Fellahi et al., 2013*), pearl millet (*Velu et al., 2011b*; *Govindaraj et al., 2013*), and maize (*Long, Bänziger & Smith, 2004*; *Chen et al., 2007*) suggesting that the grain micronutrient contents are mainly under additive genetic action.

Significant SCA effect for any character may not have significant GCA effects for parentage cross combination (*Chaudhari & Patel, 2014b*). For example, parents of H15 hybrids had significant negative GCA effects, and themselves had a significant positive SCA effect for yield characters in $E_1$. Here, no significant SCA effects were determined in all hybrids for estimated characters. It is recommended that these characters be less than the maximum mean of parents.

In the reverse case, parents with significant GCA effects did result in the best crosses with significant SCA like parents of H2, H3, and H26 which showed positive GCA effects, and crosses were observed with positive SCA effects for GYP in normal conditions.

This remark could be assigned to unlike combo of dominant and recessive genes from particular parents; it far justifies the non-additive gene action (additive × dominance, dominance × dominance, and epistatic interactions). The cross H3 was observed with the highest significant positive SCA effect for GYP in normal conditions; GYP, PC, and grain Fe content in late conditions. For grain Fe and Zn content, the hybrid H43 had a significant positive SCA effect in both environments. Generally, hybrids with high SCA effects are recommended for heterosis breeding. Thus, these hybrids could be exploited in the breeding program to select useful segregants in succeeding generations and consequently, it would be valuable to utilize them to enhance grain yield and micronutrients.

Therefore, the GCA effects of parents are insufficient to anticipate the SCA effects of respective hybrids. Conversely, a significant association between SCA effects and heterosis was noticed for all studied characters in both environments. In common, heterosis breeding targeted for particular traits based on the GCA effects of parents can be taken as the best cross combinations.

These hybrids involved at least one good general combiner parent and would produce transgressive segregants. Although, for full exploitation of existing genetic variance in these hybrids, intermating of best plants in early segregating generations would be useful to assemble the best populace of parents with high micronutrients and grain yield. However, hybrids combo with high SCA estimates can be exploited in heterosis breeding. Additive gene action act for GCA, and non-additive gene action compute SCA.

In the late condition, a positive correlation between grain Fe and Zn content was noticed in parents and the hybrids. Similar associations among these grain micronutrient content have been noted formerly in rice (*Stangoulis et al., 2007*; *Anandan et al., 2011*), maize (*Arnold, Bauman & Aycock, 1977*), wheat (*Garvin, Welch & Finley, 2006*; *Velu et al., 2011a*), sorghum (*Kumar et al., 2010*; *2013*), pearl millet (*Govindaraj et al., 2013*; *Velu et al., 2008*; *Rai, Govindaraj & Rao, 2012*; *Kanatti et al., 2014*), and finger millet (*Upadhyaya et al., 2011*). To the best of our knowledge, this is the first report evaluating parental wheat lines and hybrids under normal and late sown conditions for grain protein and micronutrient content.

## CONCLUSION

This study elucidated variation in GYP and protein, Fe, and Zn content under normal and late sown conditions. Previously, protein and micronutrient contents have rarely been estimated under normal and late sown conditions. The present findings indicate that the hybrids exhibited higher mean micronutrient and protein contents under late sown condition than normal. The study paved the way for further improvement of GYP and protein, Fe, and Zn contents under late sown conditions. However, it will be necessary to introduce high Fe and Zn densities into both parental lines in order to breed hybrids, and the use of genomics techniques may greatly speed up this procedure. Parental lines of potential hybrids with high general combining ability (GCA) could be successfully selected based on their performance, increasing the breeding efficiency, according to a highly significant and positive correlation between their performance and their GCA for both Fe and Zn densities. Further research is necessary because of the direct impact these findings have on the effectiveness of breeding high-yielding hybrids with high levels of Fe and Zn densities. Inbred lines show no correlation between Fe and Zn densities and grain yield, but hybrids show a significant negative (though low) correlation.

## ACKNOWLEDGEMENTS

We would like to thank Anand Agricultural University for providing farm facilities, micronutrient & biochemical analysis.

### Funding

The authors received no funding for this work.

### Competing Interests

Sushil Kumar is an Academic Editor for PeerJ. The authors declare that they have no competing interests.

### Author Contributions

- Gita R. Chaudhari conceived and designed the experiments, performed the experiments, analyzed the data, prepared figures and/or tables, authored or reviewed drafts of the article, and approved the final draft.

- D. A. Patel conceived and designed the experiments, analyzed the data, authored or reviewed drafts of the article, and approved the final draft.
- D. J. Parmar analyzed the data, authored or reviewed drafts of the article, and approved the final draft.
- K. C. Patel analyzed the data, authored or reviewed drafts of the article, and approved the final draft.
- Sushil Kumar performed the experiments, authored or reviewed drafts of the article, and approved the final draft.

## Data Availability

The raw measurements are available in the Supplemental Files.

## Supplemental Information

Supplemental information for this article can be found online at http://dx.doi.org/10.7717/peerj.14971#supplemental-information.

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
