# Peer review of "Combining ability, heterosis and performance of grain yield and content of Fe, Zn and protein in bread wheat under normal and late sowing conditions"

_PeerJ, doi:10.7717/peerj.14971_

## Round 0.1 · original submission · Major Revisions

Dear Authors
Revise the manuscript as per the comments of the reviewers and resubmit for consideration.

Please include normal agriculture practices of crop.

·

Basic reporting

1. The introduction section should be drafted so as there is a continuity in the flow of data.
2. The unit for data reported needs to be provided in table 2.
3. The full form of the abbreviations used needs to be provided at the first mentioned.
4. The discussion could be further enhanced
5. Similar research has been reported in 2018, highlight the novelty of the study.

Experimental design

1. The study may be carried out for multi years at multilocation trials.

Validity of the findings

1. The study can be further validated by multi year study for reliability of stable identified lines.

Reviewer 2 ·

Basic reporting

The manuscript provides a fairly robust dataset on combining ability, heterosis, and gene action of grain yield and content of Fe, Zn, and protein in bread wheat under normal and late sowing conditions. The authors put effort into crossing ten wheat genotypes and developed F1 crosses which were evaluated under normal and late sowing conditions. I suggest accepting the manuscript following major revision.
The manuscript needs major English editing.
The title should be improved, it could be “Combining ability, heterosis and gene action of grain yield and content of Fe, Zn and protein in bread wheat under normal and late sowing conditions”
The introduction needs to be improved, the hypothesis needs to be clarified. The importance of Fe and Ze content in wheat grain, the importance of statistical analysis (combining ability, heterosis, and gene action), the negative impact of late sowing need to be extended and improved.
Ms&Ms
More details should be added on the agricultural practices, N, P, and K fertilization rates, and water irrigation amount…..
Lines 111-114 should not be presented under the subtitle “Statistical analysis” but should be presented earlier above under subtitle “Description of the experimental site”
The result section needs to be improved and gene action is not presented which should be presented. Supplementary table 1 is important, it should be presented in the main text.
The obtained results should be discussed better and citations should be updated.

Experimental design

Adequite

Validity of the findings

Adequite

---

## Round 0.2 · Major Revisions

Revise the manuscript as per the comments of the reviewers and resubmit for consideration.

·

Basic reporting

1. The previous comment has not been addressed “The introduction section should be drafted so as there is a continuity in the flow of data.” Redraft the starting sentence of the paragraph “Effects of temperature and day length are mainly related to different sowing dates. In central India, short duration….” To bring a continuity rather than merging the paragraphs.
2. The novelty needs to be highlighted in the MS.
3. The previous comment “The unit for data reported needs to be provided in table 2” has been reported to be addressed by the authors in the rebuttal letter but the changes are not seen in the table 2. Rectify it.
4. Many old references are cited in the MS which may be replaced with recent ones.

Experimental design

Adequate

Validity of the findings

Adequate

Reviewer 2 ·

Basic reporting

The authors have addressed previous concerns in the earlier revision. Just still the units should be added in Table 2 as well as LSD.

Experimental design

Adequite

Validity of the findings

Adequite

---

## Round 0.3 · Minor Revisions

Some minor revisions are required before acceptance.

·

Basic reporting

1. New references needs to be added to replace the older references.
2. The references are not properly cited. Incomplete references needs to be rectified as for e.g. reference no. 25, 32, 34, 35. This needs to be rectified throughout the MS.

Experimental design

Adequate

Validity of the findings

Adequate

Reviewer 2 ·

Basic reporting

Adequate

Experimental design

Adequate

Validity of the findings

Adequate

Additional comments

The authors have addressed previous concerns in the earlier revision.

---

## Round 0.4 · accepted · Accept

All the comments has been resolved properly